# A Paper-Based Analytical Device for Analysis of Paraquat in Urine and Its Validation with Optical-Based Approaches

**DOI:** 10.3390/diagnostics11010006

**Published:** 2020-12-22

**Authors:** Tse-Yao Wang, Yi-Tzu Lee, Hsien-Yi Chen, Cheng-Hao Ko, Chi-Tsung Hong, Jyun-Wei Wen, Tzung-Hai Yen, Chao-Min Cheng

**Affiliations:** 1Department of Emergency Medicine, Taipei Veterans General Hospital, Taipei 112, Taiwan; tseyao85@gmail.com (T.-Y.W.); s851009@yahoo.com.tw (Y.-T.L.); 2School of Medicine, National Yang-Ming University, Taipei 112, Taiwan; 3Department of Emergency Medicine, Chang Gung Memorial Hospital, Taoyuan 333, Taiwan; hshychen@gmail.com; 4College of Medicine, Chang Gung University, Taoyuan 333, Taiwan; 5Graduate Institute of Automation and Control, National Taiwan University of Science and Technology, Taipei 106, Taiwan; dr.ko.chenghao@gmail.com; 6Spectrochip Inc., Hsinchu 300, Taiwan; cthong23@gmail.com (C.-T.H.); wenjyunwei@gmail.com (J.-W.W.); 7Institute of Biomedical Engineering, National Tsing Hua University, Hsinchu 300, Taiwan; 8Department of Nephrology, Clinical Poison Centre, Kidney Research Centre, Centre for Tissue Engineering, Chang Gung Memorial Hospital and Chang Gung University, Taoyuan 333, Taiwan

**Keywords:** paraquat poisoning, paper-based analytical device, point-of-care testing

## Abstract

Paraquat is a highly toxic herbicide. Paraquat poisoning is often fatal and is an important public health threat in many places. The quick identification and timely initiation of treatment based on timely analysis of the paraquat concentration in urine/serum could improve the prognosis for patients. However, current paraquat concentration measurements are time-consuming and difficult to implement due to the expensive and bulky equipment required. To address these practical challenges, paper-based devices have emerged as alternative diagnostic tools for improving point-of-care testing. In this study, we demonstrate the successful use of a paper-based analytical device for the accurate detection of urine paraquat concentration. The developed paper-based analytical device employs colorimetric paraquat concentration measurements. The R^2^ value for the urine paraquat standard curve was 0.9989, with a dynamic range of 0–100 ppm. The limit of detection was 3.01 ppm. Two other optical-based approaches, Spectrochip and NanoDrop, were used for comparison. The results suggest that the developed paper-based analytical device is comparable to other colorimetric measurements, as determined by Bland–Altman analysis. The device was clinically validated using urine from six paraquat-poisoned patients. The results prove that the developed paper-based analytical device is accurate, easy-to-use, and efficient for urine paraquat concentration measurement, and may enable physicians to improve clinical management.

## 1. Introduction

Paraquat (*N,N′-*dimethyl-4,4′-bipyridinium dichloride) is one of the most widely used herbicides in the world. Introduced for agricultural use in the 1960s, this non-selective herbicide became a low-cost, readily accessible, and prevalent product globally. Paraquat is a highly toxic agent and paraquat ingestion, whether intentional or accidental, is frequently fatal. Paraquat has been prohibited in many countries due to its lethality, but paraquat poisoning remains an important public health threat in many regions [1]. A mortality rate of 60–70% and an estimated annual incidence of 2000 toxic ingestions have been reported in some Asian countries [2,3].

Paraquat toxicity occurs mainly via a mechanism involving the production of large amounts of toxic free radicals. These toxic free radicals cause lipid peroxidation in the cell membrane, exhaust nicotinamide adenine dinucleotide phosphate (NADPH), and lead to cell death. The most characteristic feature of paraquat poisoning is lung damage. These effects occur during the first several hours following paraquat ingestion. Production of toxic free radicals and subsequent inflammation lead to lung fibrosis and respiratory failure. Paraquat also attacks other organs. In larger doses, paraquat ingestion can cause multi-organ failure and death [4,5].

The clinical prognosis for acute paraquat poisoning is dose dependent. Paraquat concentrations in plasma and urine are strongly correlated with the prognosis. The ingestion of large doses of paraquat may lead to death due to multi-organ failure and shock within only a few days [1,5]. The severity index of paraquat poisoning (SIPP) is determined by multiplying the serum paraquat concentration at the time of patient admission (ppm) with the time to treatment (h) [6]. The SIPP is correlated with the prognosis, where SIPP values over 10 indicate a high probability of death [7]. In addition, urine concentration has important diagnostic value and carries prognostic implication. Patients with semiquantitative urine test results showing darker than navy blue (>10 ppm) in the initial 24 h have a high probability of death [5,8]. This semiquantitative urine test using the sodium dithionite reaction is also the pioneer of point-of-care test application in clinical toxicology. A recent study suggested that initial urine paraquat concentration below 32.2 ppm significantly correlated with a higher 28-day survival rate [9].

Various methods of measuring paraquat concentration have been proposed, including high-performance liquid chromatography (HPLC)/mass spectrometry (MS), gas chromatography (GC)/MS, and photometry coupled with a sodium dithionite assay [1,10,11]. However, these methods are time-consuming. Furthermore, the bulky instrumentation and high examination cost make the widespread implementation and accessibility of these methods difficult. The sodium dithionite urine assay may be conducted easily and rapidly, but this semiquantitative method may not be precise enough to guide treatment. Thus, new point-of-care tests, which are easily performed, less invasive, and timesaving, are warranted for better acute paraquat management.

In recent years, paper-based analytical devices have become increasingly well-developed. Microfluidic paper-based analytical devices were introduced in 2007 [12]. The properties of the paper-form micro-channel networks can facilitate easy sample handling and quantitative analysis. Several possible detection methods have been introduced, including colorimetric detection, electrochemical detection, fluorescence detection, chemiluminescence detection, electrochemiluminescence (ECL) detection, and photoelectrochemical detection [13]. A wide variety of paper-based analytical devices have recently gained prominence in the fields of medicine, healthcare, and environmental monitoring [14,15]. The advantages of paper-based analytical devices are their relatively low-cost, simple operation, speed, and portability. Various applications were introduced recently, including measurement of hematocrit, creatinine, glucose, electrolytes, tumor markers, virus, and so on [13,14,15]. In the toxicology field, our team and other studies also presented the measurement of serum paraquat and organophosphates using paper-based analytical devices [16,17]. Paper-based analytical devices offer two important futuristic applications. The first is as a diagnostic tool in a low-infrastructure environment. The other is for alternative point-of-care testing (POCT) prior to time-consuming, sophisticated measurement.

This study demonstrates the novel design of a paper-based analytical device for detecting paraquat poisoning. The developed device is compared with other commercially available colorimetric detection methods, i.e., Spectrochip and NanoDrop. The device is also subjected to clinical validation using urine samples from six paraquat-poisoning patients.

## 2. Methods

### 2.1. Chemicals

Paraquat dichloride hydrate (Sigma Aldrich, St. Louis, MO, USA), sodium hydroxide (NaOH) (Sigma Aldrich, St. Louis, MO, USA), sodium dithionite (85%, Sigma Aldrich, St. Louis, MO, USA), phosphate buffered saline (PBS) (tablet, 85%, Sigma Aldrich, St. Louis, MO, USA), and Whatman qualitative filter paper, No. 1 (GE Healthcare Life Science; No. 1001-150) were used in the experiments.

### 2.2. Paper-Based Analytical Device

The paper-based analytical device was prepared as 96-well plates on paper substrates. Specifically, we designed a 96-well template pattern (8 × 12 circle array) using Microsoft Office software. The diameter of each well was 0.4 cm. We then fabricated the paper-based microzone plates using wax-based printing technology on to paper (Whatman qualitative filter paper No. 1). We used a wax printer (Xerox Phaser 8650 N color printer, Xerox Corporation, Norwalk, CT, USA) for printing. Afterward, the paper device with the wax printed wells was heated by an oven for 5 min at 105 °C. In this step, the wax melted and allowed it to penetrate through the highly permeable paper. Finally, the hydrophilic chemical reaction area with complete, well-defined hydrophobic wax barriers was formed. The chemical reaction area impregnated with reactive reagents that produced a color change following the application of the patient test samples. The chemical reaction area was also the detection test zone, which allows us to measure the color intensity change. The schematic diagram of our paper-based analytical device is presented as Figure 1.

### 2.3. Colorimetric Assay

The mechanism of the colorimetric assay is based on the chemical reaction of paraquat and sodium dithionite, which results in the formation of a blue radical ion. The paper-based colorimetric assays were conducted as follows: (1) 5 μL of 5 N NaOH and 2 μL of 20% (*w/v*) sodium dithionite were placed onto each detection zone of the paper-based 96-well plate; (2) 8 μL of the experimental sample was subsequently added to each well before the reagents dried out. Standard samples of various concentrations (paraquat standard solutions dissolved in PBS at 0, 5, 10, 25, 50 ppm) and patient samples (diluted in PBS) were applied to the wells.

The chemical reaction was conducted under ambient conditions. Based on our kinetics study under ambient conditions (Appendix A), we set the reaction time at 10 min to obtain the best detection results.

The reaction results were subsequently preserved by a digital camera (EOS 5D Mark III, Canon, Japan). The paper-based analytical device was implemented on a white base background. The digital camera was set to auto-focus mode. We captured the image perpendicularly in ambient light.

The resulting data were then analyzed using Image J software (Version 2.0.0, National Institute of Health). The region of interest (ROI) of each detection zone was selected as a circle with a diameter of 0.38 cm (95% of the diameter of each well) to reduce the interference of reflection. We minimized other interferences of inhomogeneous color intensity by maintaining an equal condition in each measurement. Then, we analyzed the RGB color values of each detection zone before and after testing, where R, G, and B are the red, green, and blue coordinates. The ΔRGB value is the mean intensity difference in RGB analysis after 10 min reaction according to the below equation.
ΔRGB = ((ΔR)^2^ + (ΔG)^2^ + (ΔB)^2^)^1/2^

All samples were analyzed in triplicate. The standard deviation, limit of detection, and limit of quantification of the developed paper-based device were also calculated.

### 2.4. Spectrochip and NanoDrop

Spectral analysis instruments have been used for chemical and biochemical analyses, immunoassays, biosensors, etc. Two spectral detection devices were compared with the developed device, i.e., a spectrophotometer (NanoDrop, Thermo Fisher Scientific, Waltham, MA, USA) and a chip-based spectrometric device (Spectrochip, Spectrochip Inc., Hsinchu County, Taiwan). The Spectrochip device employs innovative micro-grating technology to create a palm-size portable point-of-care testing (POCT) device for biomedical spectral analysis [18,19]. The spectral range of this device is from 300 to 1100 nm with a selectable spectral resolution between 5 and 15 nm.

For comparative evaluation of the test samples, 20% (*w/v*) sodium dithionite solution was diluted in 1 N NaOH, 500 μL was loaded into each test cuvette, and the paraquat samples were added. The paraquat samples included paraquat standard solutions (0, 5, 10, 50, 150 ppm) and patient urine samples. The chemical reaction resulted in a color change from colorless to blue, and the color data were detected using a traditional spectrophotometer (NanoDrop, Thermo Fisher Scientific) and a new spectrometric device (Spectrochip), respectively.

### 2.5. Statistical Analysis

The student’s *t*-test was used to evaluate the detection results with standard values from clinical reports. A value of *p* < 0.05 was considered statistically significant.

### 2.6. Clinical Validation and Collection of Patient URINE Samples

Clinical urine samples were collected at Chang Gung Memorial Hospital, Linkou Medical Centre, Taiwan. This study complied with the guidelines of the Declaration of Helsinki. Informed consent was obtained from all participants. The study protocol was approved by the institutional review board of Linkou Chang Gung Memorial Hospital on 25 September 2018 (IRB No. 201801259B0).

A total of six urine samples were collected from individual patients suffering from paraquat poisoning. Clinical validation of the developed device was conducted by comparing the paraquat concentration determined using the other devices.

## 3. Results

### 3.1. Paraquat Detection with the Paper-Based Analytical Device

After reaction of the paraquat sample in the paper-based analytical device, the color intensity was measured and the *ΔRGB* value (*RGB* = red, green, blue color coordinates) was calculated as described in the Methods section. Table 1 shows the results of triplicate tests for the standard paraquat solution using the paper-based analytical device. The limit of paraquat detection and limit of paraquat quantification were 3.01 and 10.02 (ppm), respectively.

We then established the calibration function and standard curve using six different concentrations of paraquat (0, 5, 10, 25, 50, and 100 ppm). Figure 2 illustrates a paraquat standard curve using data from the paper-based analytical device, with a concentration range of 0–100 ppm. The regression function was obtained and the correlation coefficient (*R*^2^ value) in our system was 0.9989.

### 3.2. Paraquat Detection with Spectrochip and NanoDrop

Two spectral detection devices were compared with the developed device, i.e., a spectrophotometer (NanoDrop, Thermo Fisher Scientific) and a chip-based spectrometric device (Spectrochip, Spectrochip Inc.). Five different paraquat concentrations (5, 10, 50, 100, 150 ppm) were measured using Spectrochip and NanoDrop respectively.

The transmission of white light from a light-emitting diode (LED) was measured with Spectrochip. The transmittance data were converted to absorbance data by using the relation: *A* = −log (*T*), where *A* and *T* are the absorbance and transmittance, respectively. The calibration function and standard curve were established (Figure 3).

The absorption of white light from the LED was also measured with a common UV/visible spectrometer (NanoDrop); the absorbance increased as the color intensity increased. The calibration function and standard curve were established as shown in Figure 4.

The spectra of serial concentrations of the paraquat samples were measured using both devices. A transmittance peak at 600.405 nm was noted in the Spectrochip data. An absorbance peak at 603 nm was detected via NanoDrop. Two calibration curves were constructed based on the transmittance of paraquat at 600.405 nm for solutions of differing concentrations and based on the absorbance at 603 nm to create an exponential plot and sigmoidal plot, respectively.

### 3.3. Comparison of Paraquat Detection via Different Methods Using Bland–Altman Analysis

The results from the developed paper-based analytical device were compared to the results from the Spectrochip and NanoDrop systems using Bland–Altman analysis and were found to be similar (Figure 5). Bland–Altman analysis of the results from the paper-based analytical device versus the results from Spectrochip revealed a mean difference of −558.045 (red line); the dotted lines represent the 95% limits of agreement (Figure 5a). Bland–Altman analysis was also used to compare the data from the paper-based analytical device with that from NanoDrop (Figure 5b) and to compare Spectrochip with NanoDrop (Figure 5c), where the mean differences are respectively −431.068 and 126.977. The agreement across the results from all approaches was within the 95% confidence limit. The Bland–Altman analysis validation indicates that the developed paper-based analytical device provides colorimetric detection power comparable to that of the traditional colorimetric methods.

### 3.4. Clinical Validation Using Six Patient Urine Samples

Clinical validation of the developed paper-based analytical device was conducted by comparing the results to the measurements using the other clinical devices. Table 2 summarizes the clinical data from six patients with paraquat poisoning. The patients were 24–94 years old, and all were male. All patients attempted suicide by drinking 24% paraquat (Gramoxone, Syngenta, Taiwan), and were admitted to Chang Gung Memorial Hospital within 1.5–5.0 h. Their blood paraquat concentrations were 26.4 ppm, 142.4 ppm, >8.0 ppm, 4.9 ppm, >10.0 ppm, and >10.0 ppm, respectively. The severity indexes of paraquat poisoning (SIPP) were greater than 10. As shown in Table 2, all patients developed serious medical complications and expired despite intensive resuscitation efforts.

Figure 6 displays the analytical concentrations of paraquat in the patient urine samples from the Spectrochip and NanoDrop methods. Table 3 shows the sample data for all methods and demonstrates that the developed paper-based analytical device is as accurate and efficient as spectrophotometric analysis.

## 4. Discussion

This study demonstrates a paper-based analytical device that can be used for detection of paraquat poisoning. The paper-based analytical device employed the chemical reaction of paraquat and sodium dithionite in alkaline solution. The colorimetric method was used for concentration measurement. The R^2^ value for the standard curve using data from the developed paper-based analytical device was 0.9989, with a concentration range of 0–100 ppm. For paraquat detection, the limit of detection and limit of quantification were 3.01 and 10.02 (ppm), respectively. Standard curves for paraquat detection were also established by using data from the other two methods, Spectrochip and NanoDrop, for comparison. Although the two devices are commercialized, their clinical applications are limited and under-developed now, especially in the clinical toxicology field. As new diagnostics tools for paraquat detection, the standard measuring process of the two devices has not been established. In addition, the devices are not clinically available in hospitals. The R^2^ values for the standard curves of the data from Spectrochip and NanoDrop were 0.9888 and 0.9941, respectively.

Furthermore, the developed paper-based analytical device was clinically validated by using patient samples. The measurement results from the developed paper-based analytical device, Spectrochip, and NanoDrop were compared using Bland–Altman analysis. The developed paper-based analytical device showed consistency with Spectrochip and NanoDrop, as determined by Bland–Altman analysis. Our experiment indicates that urine paraquat analysis using the developed paper-based analytical device is comparable to that achievable with other traditional colorimetric methods, which are more complicated and time-consuming.

Acute paraquat intoxication can be fatal and it remains a clinical challenge. Although there is no gold-standard treatment for paraquat intoxication, except supportive care, recent evidence suggests that timely hemoperfusion is beneficial for resolving acute paraquat poisoning. Hemoperfusion conducted within four to five hours following paraquat ingestion, combined with pulse therapy, could reduce the mortality [20]. A recent meta-analysis also supports the potential benefit of hemoperfusion [21]. Therefore, quick diagnosis of acute paraquat poisoning and determination of the degree of paraquat exposure are the cornerstones to effective management. Because the time elapsed before hemoperfusion is associated with the prognosis, early recognition of paraquat poisoning and quick initiation of hemoperfusion and other treatments are important. However, the toxidrome of paraquat intoxication is non-remarkable, and the patient history is not always reliable. Therefore, measurements of plasma or urine paraquat concentrations are the diagnostic methods of choice for acute paraquat poisoning, and timely measurement is key to guiding effective care [1,5,7,8,9,10].

We developed a paper-based analytical device employing a colorimetric method. The whole detection process is simple and the results are of relatively good quality. The accuracy of the paper-based analytical device is superior to that of quick urine sodium dithionite assays currently in clinical use, and the measurement requires only a short time for completion. In addition to its utility as a diagnostic tool, this time-saving and simple detection device could be used for easy follow-up care and may allow physicians to precisely decide on a treatment regimen based on timely concentration analysis. Paper-based analytical devices can also be developed to provide a platform for simultaneous sample analyses, which further underscores their utility, efficiency, and cost-saving features, and supports their potential for commercial product development. Our study proves that paper-based analytical devices can improve point-of-care testing and may improve acute paraquat poisoning treatment.

This paper-based analytical device was developed for the analysis of paraquat in urine. Previously, our laboratory developed a paper-based analytical device designed for the rapid assay of paraquat in human serum [16]. The previous device employs a colorimetric sodium dithionite assay or an ascorbic acid assay, which can be used to determine the paraquat concentration in human serum in less than 10 min. In another study [17], a 2-in-1 paper-based analytical device was developed to simultaneously measure paraquat and creatinine concentrations in human serum. This 2-in-1 device is inexpensive, simple, and provides rapid detection of paraquat while assessing renal function. The development of a tool for urine paraquat testing, however, provides a simpler approach for clinical diagnosis of paraquat poisoning, which increases the accessibility to diagnostics and facilitates rapid treatment. In the case of criminal investigation, urine testing is the most commonly used test for illicit drug screening. The advantages of urine testing include its ease of use and non-invasive nature. Compared with blood tests, the urine test can be easily performed by nonmedical or paramedical professionals at the time and place of patient care. Furthermore, the urine test can disclose the presence of the drug in the human body after its clinical effects or blood concentrations have worn off. For example, the detection times for amphetamine are 2–4 days and 12 h in urine and plasma, respectively [22]. In addition to diagnostic value, the urine paraquat concentration also carries prognostic implication. In a study of 194 paraquat patients, Liu et al. [9] reported that the areas under the curve values of urine paraquat concentration for predicting early and delayed mortality were 0.890 and 0.764, respectively. The data suggest that urine paraquat concentration could serve as a useful biomarker for predicting outcomes of patients with paraquat poisoning.

## 5. Conclusions

We successfully developed a paper-based analytical device for urine paraquat concentration measurement. The analytical device can simplify measurement procedures and provide quantifiable colorimetric results. Our paper-based analytical device is an ideal point-of-care tool that is demonstrably accurate, less-invasive, easy-to-use, and time- and cost-effective. This device could be an efficient tool and enable physicians to improve acute paraquat intoxication management.

## Figures and Tables

**Figure 1 diagnostics-11-00006-f001:**
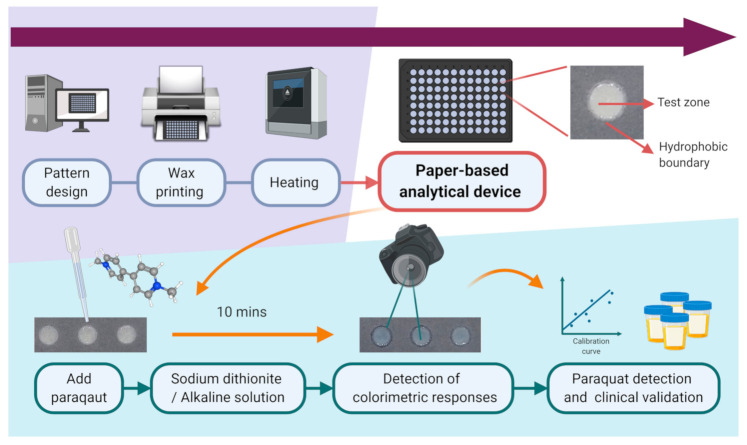
The schematic diagram of our paper-based analytical device for urine paraquat detection.

**Figure 2 diagnostics-11-00006-f002:**
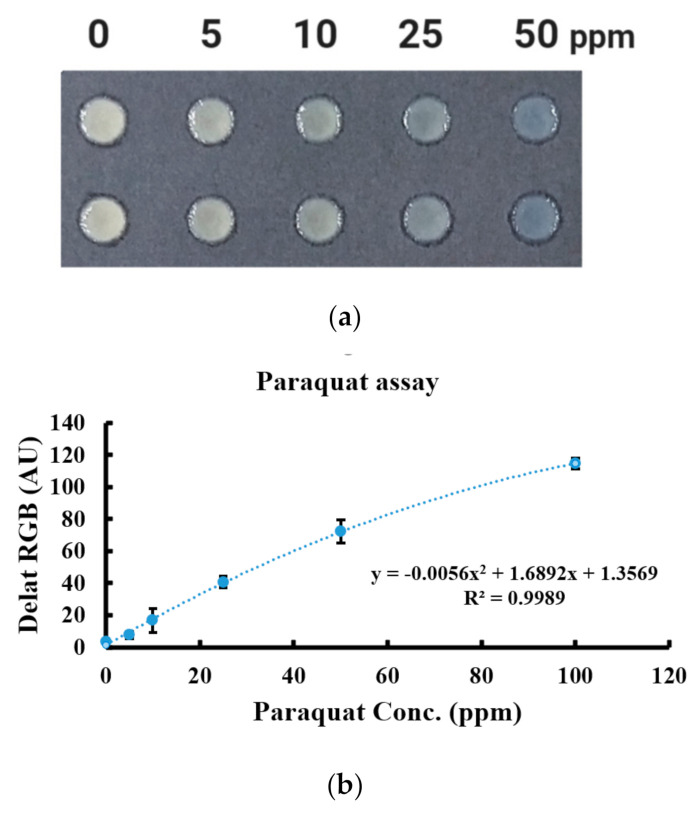
Results from developed paper-based assay for paraquat detection. (**a**) The reaction results of serial concentrations of paraquat. (**b**) Calibration curve for paraquat using the paper-based analytical device.

**Figure 3 diagnostics-11-00006-f003:**
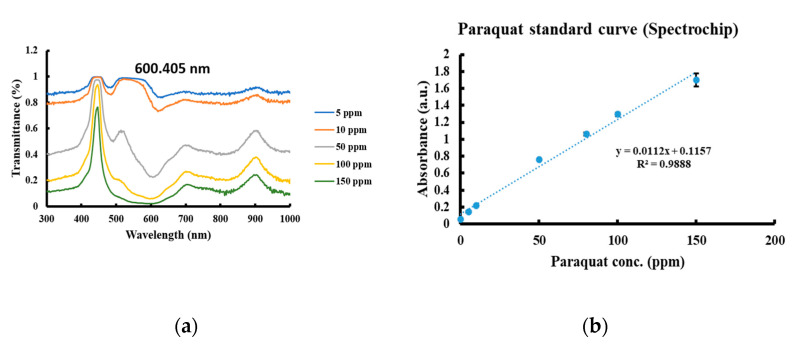
Paraquat determination using Spectrochip measurements. (**a**) Absorption spectrum of serial concentrations of paraquat in PBS. (**b**) Paraquat standard curve.

**Figure 4 diagnostics-11-00006-f004:**
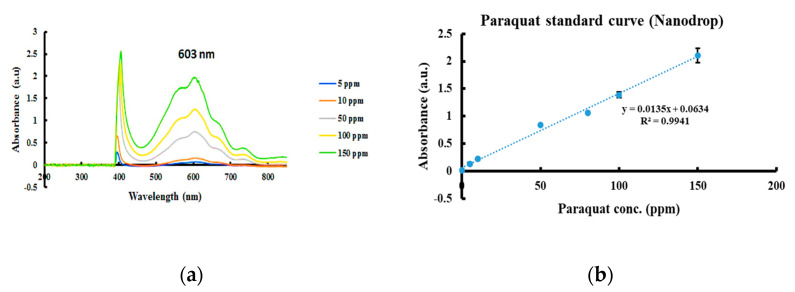
Paraquat determination using NanoDrop measurement. (**a**) Absorption spectrum of serial concentrations of paraquat in PBS. (**b**) Paraquat standard curve.

**Figure 5 diagnostics-11-00006-f005:**
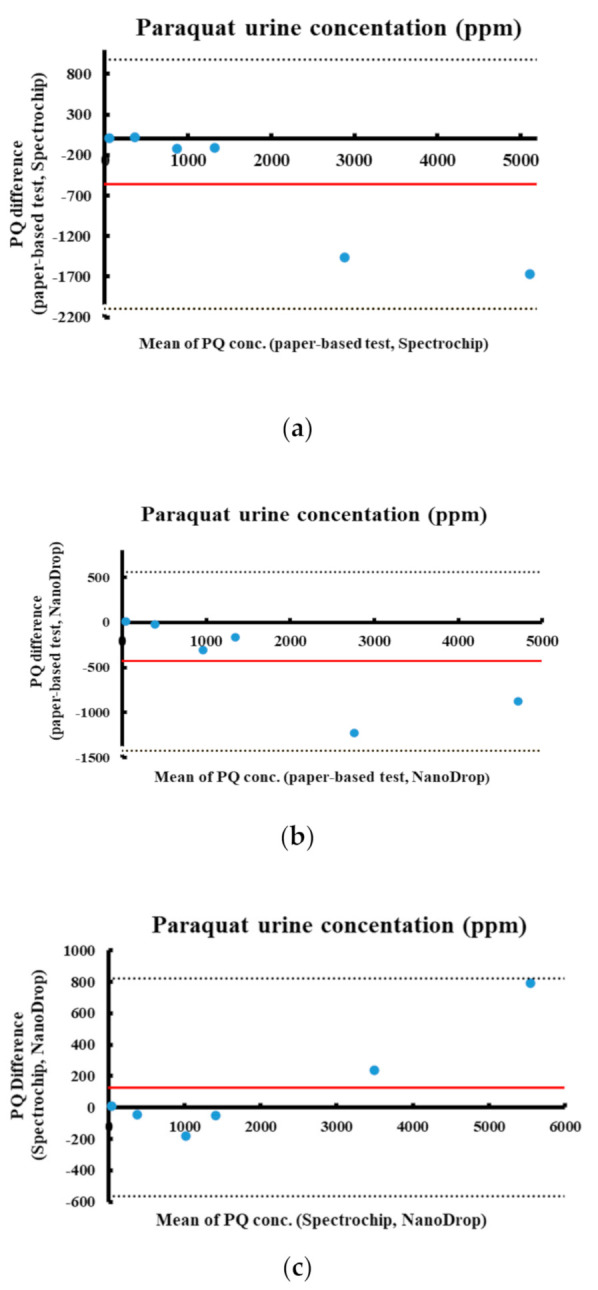
Bland–Altman plots of paraquat measurements. (**a**) Comparison of paper-based method and Spectrochip. (**b**) Comparison of paper-based method and NanoDrop. (**c**) Comparison of Spectrochip and NanoDrop. Dotted lines represent the 95% limits of agreement. Red line is the average difference between the methods.

**Figure 6 diagnostics-11-00006-f006:**
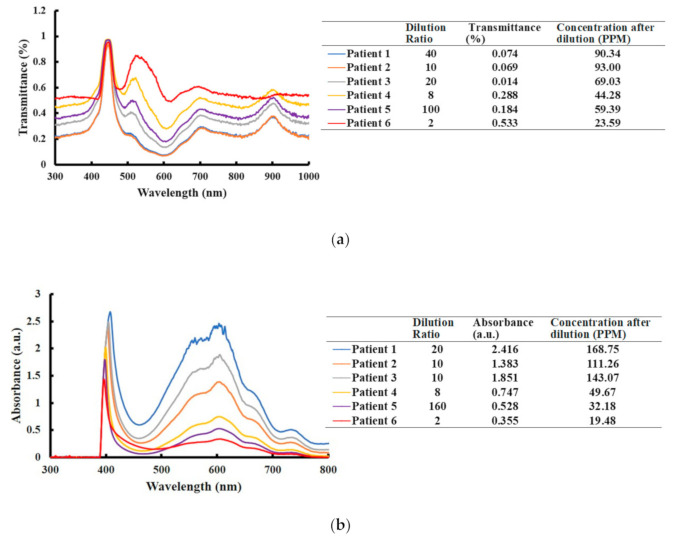
(**a**) Transmission spectrum of patient urine samples diluted in PBS as measured by Spectrochip. (**b**) Absorption spectrum of patient urine samples diluted in PBS as measured by NanoDrop.

**Table 1 diagnostics-11-00006-t001:** Results from developed paper-based device for standard paraquat solution detection.

Delta RGB *	0 ppm	5 ppm	10 ppm	25 ppm	50 ppm	100 ppm
Test 1	3.26457	6.07037	8.28258	42.7104	66.5518	111.381
Test 2	5.25096	6.65480	20.7853	36.3824	69.7230	118.404
Test 3	2.23013	9.92608	21.4377	42.8390	80.4217	114.011
Average	3.58189	7.55042	16.8352	40.6439	72.2322	114.599
Standard deviation	1.53521	2.07803	7.41396	3.69117	7.26746	3.54797
Limit of detection: 3.01 ppm						
Limit of quantification: 10.02 ppm						

* ΔRGB = ((ΔR)^2^ + (ΔG)^2^ + (ΔB)^2^) ^1/2^.

**Table 2 diagnostics-11-00006-t002:** Clinical data for patients with paraquat poisoning.

Patient Number	1	2	3	4	5	6
Age	24	26	49	63	94	76
Sex	Male	Male	Male	Male	Male	Male
Time elapsed between paraquat ingestion and hospital arrival, (h)	1.5	2.0	4.0	5.0	3.0	3.0
Blood paraquat level, (ppm)	26.4	142.4	>8.0	4.9	>10.0	>10.0
SIPP, (ppm h)	39.6	284.8	>32.0	24.5	>30.0	>30.0
Treatment	Charcoal hemoperfusion, glucocorticoid/cyclophosphamide pulse therapies	Charcoal hemoperfusion	Death before treatment	Charcoal hemoperfusion, glucocorticoid/cyclophosphamide pulse therapies	Death before treatment	Death before treatment
Duration of hospitalization, (day)	2	0.5	0.1	2	0.2	0.1
Outcome	Dead	Dead	Dead	Dead	Dead	Dead

Note: SIPP, severity index of paraquat poisoning. The SIPP was derived from the product of the plasma paraquat level in ppm and the time elapsed between paraquat ingestion and hospital arrival in hours [6].

**Table 3 diagnostics-11-00006-t003:** Analytical methods for paraquat determination in urine samples.

Patient Number	1	2	3	4	5	6
Quick urine sodium dithionite test (ppm)	>50	>50	>50	>50	>50	>50
Paper-based (ppm)	2145.6	804.0	1267.5	372.8	4275.0	51.5
Spectrochip (ppm)	3613.6	930.0	1380.6	354.2	5939.0	47.2
NanoDrop (ppm)	3375.0	1112.6	1430.7	397.4	5148.2	39.0

## Data Availability

The data that support the findings of this study are available from the corresponding author upon reasonable request. The data are not publicly available due to it contain patient information.

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
