# Peer review of "A Paper-Based Analytical Device for Analysis of Paraquat in Urine and Its Validation with Optical-Based Approaches"

_diagnostics, 2020, doi:10.3390/diagnostics11010006_

Round 1
Reviewer 1 Report
The paper-based chip for analysis of paraquat in urine. Paraquat is a awful and highly toxic herbicide. People with large ingestions of paraquat are not likely to survive.(US CDC) This research is very important to improve the prognosis for patients.
Strength
1. The urine paraquat concentration carries prognostic implication and is very important.
2. 10min reaction time.
3. Utility, efficiency, and cost-saving features.
Weakness
1. There are several kinds of paraquat-poisoned patients, including direct drinking of pesticide (suicide or carelessly misdrinking) and poisoning in using process( body exposure lasts for a long time).
The patient who is lastes exposure topical and inhalation for a long time will be also paraquat poisoning. After exposure topical and inhalation, detecting the paraquat concentration in urine in several hours later is more important?
So, there are two different concentration ranges of paraquat in poisoned patient's urine. Is the urine concentration of patient poisoning in using process lower than direct drinking? And resolution of the system is enough in low concentraton?
2. The paraquat concentration in serum is becoming higher than urine earlier. Why urine, but not serum?
3. There are reflecting light on each wells in figure 2(a). Because of each well's location is different, the reflecting light area is also different. Is the reflecting light area interference measuring result?
4. Is the ΔRGB calculating value include only blue color coordinate or three color coordinates?
Author Response
We greatly appreciate the time and effort the editor and reviewers have put into our paper. Below we have outlined our responses to the reviewer’s comments point by point. We hope that the following responses and the corresponding revision of the manuscript meet the editor’s and reviewers’ requirements for considering this manuscript for publication in Diagnostics.
In the following sections, we have restated the reviewer’s comments followed by our response to each comment.
- There are several kinds of paraquat-poisoned patients, including direct drinking of pesticide (suicide or carelessly misdrinking) and poisoning in using process( body exposure lasts for a long time). The patient who is lastes exposure topical and inhalation for a long time will be also paraquat poisoning. After exposure topical and inhalation, detecting the paraquat concentration in urine in several hours later is more important? So, there are two different concentration ranges of paraquat in poisoned patient's urine. Is the urine concentration of patient poisoning in using process lower than direct drinking? And resolution of the system is enough in low concentration?.
Response:
Thank you for your important comment.
The diagnosis and management of paraquat toxicity is the main focus in cases of acute intoxication. Our paper-based analytical device has been validated in acute poisoning cases. Owing to the good degradation of paraquat, relatively short bio-elimination time, and lack of serious cases, more studies are warranted to delineate chronic paraquat intoxication.
Paraquat can be absorbed via oral ingestion, inhalation, dermal exposure, and dermal injection. The maximum serum/urine concentration and elapsed time could be affected by many factors, including solvent, co-ingestion material, and other environmental conditions. Although paraquat absorption ability varies among different absorption routes, the toxicokinetics and systemic toxicity are quite similar in these routes. The local effect of acute paraquat dermal exposure is significant; however, systemic toxicity is correlated with serum/urine concentration during oral ingestion [1].
One of the difficulties in managing acute paraquat dermal exposure is that the peak concentration may be reached later than that via oral ingestion according to the skin condition. Frequently repeated measurements are important when managing cases of suspicious dermal exposure. Our paper-based analytical device, which was developed as a point-of-care test, is suitable for this situation.
We focused on paraquat clinical intoxication and management follow-up; therefore, the strengths of our developed device are short measurement time and high availability. Nonetheless, our paper-based analytical device is not designed for paraquat exposure surveillance, which requires higher resolution.
The limit of paraquat detection in our system was 3.00688 ppm. According to previous studies, patients with semiquantitative urine test results showing darker than navy blue (>10 ppm) in the initial 24 hours have a high probability of death [1,2]. A recent study suggested that initial urine paraquat concentration below 32.2 ppm had a significantly high 28-day survival rate [3]. Our paper-based analytical device could detect acute paraquat intoxication and improve clinical management.
Reference :
- Dinis-Oliveira, R.J.; Duarte, J.A.; Sanchez-Navarro, A.; Remiao, F.; Bastos, M.L.; Carvalho, F. Paraquat poisonings: mechanisms of lung toxicity, clinical features, and treatment. Crit Rev Toxicol 2008, 38, 13-71, doi:10.1080/10408440701669959
- Scherrmann, J.M.; Houze, P.; Bismuth, C.; Bourdon, R. Prognostic value of plasma and urine paraquat concentration. Hum Toxicol 1987, 6, 91-93, doi:10.1177/096032718700600116.
- Liu, X.W.; Ma, T.; Li, L.L.; Qu, B.; Liu, Z. Predictive values of urine paraquat concentration, dose of poison, arterial blood lactate and APACHE II score in the prognosis of patients with acute paraquat poisoning. Exp Ther Med 2017, 14, 79-86, doi:10.3892/etm.2017.4463.
- The paraquat concentration in serum is becoming higher than urine earlier. Why urine, but not serum?
Response:
In acute paraquat intoxication, both serum and urine concentrations have diagnostic value and carry prognostic implications. In the history of clinical toxicology, the most important and powerful tool for acute paraquat poisoning is semiquantitative urine test using sodium dithionite reaction. A study published in the 1980s was the pioneer of point-of-care test (POCT) applications [1,2]. In this article and recent studies, urine paraquat concentration measurement is still a high-value clinical tool in acute paraquat intoxication management [3].
In the concept of POCT, urine concentration has the following advantages: less invasive, requires fewer facilities and manpower, and can be conducted even in a pre-hospital setting by nonmedical or paramedical professionals. Therefore, following the development of a paper-based analytic device for paraquat serum concentration previously (our team has also developed a paper-based analytic device for paraquat serum concentration, which has already been published [4,5]), we worked to develop a paper-based analytic device for urine paraquat detection.
Reference :
- Dinis-Oliveira, R.J.; Duarte, J.A.; Sanchez-Navarro, A.; Remiao, F.; Bastos, M.L.; Carvalho, F. Paraquat poisonings: mechanisms of lung toxicity, clinical features, and treatment. Crit Rev Toxicol 2008, 38, 13-71, doi:10.1080/10408440701669959
- Scherrmann, J.M.; Houze, P.; Bismuth, C.; Bourdon, R. Prognostic value of plasma and urine paraquat concentration. Hum Toxicol 1987, 6, 91-93, doi:10.1177/096032718700600116.
- Liu, X.W.; Ma, T.; Li, L.L.; Qu, B.; Liu, Z. Predictive values of urine paraquat concentration, dose of poison, arterial blood lactate and APACHE II score in the prognosis of patients with acute paraquat poisoning. Exp Ther Med 2017, 14, 79-86, doi:10.3892/etm.2017.4463.
- Kuan, C.M.; Lin, S.T.; Yen, T.H.; Wang, Y.L.; Cheng, C.M. Paper-based diagnostic devices for clinical paraquat poisoning diagnosis. Biomicrofluidics 2016, 10, 034118, doi:10.1063/1.4953257.
- Chang, T.H.; Tung, K.H.; Gu, P.W.; Yen, T.H.; Cheng, C.M. Rapid simultaneous determination of paraquat and creatinine in human serum using a piece of paper. Micromachines (Basel) 2018, 9, doi:10.3390/mi9110586.
- There are reflecting light on each wells in figure 2(a). Because of each well's location is different, the reflecting light area is also different. Is the reflecting light area interference measuring result?
Response:
Region of interest (ROI) selection is an important issue in our measurement system. The reaction in the paper-based analytical device was captured using a digital camera perpendicularly in ambient light. Then, we analyzed the color intensity change using ImageJ software. The ROI of each well was selected as a circle with a diameter of 0.38 cm (95% of the diameter of each well) to reduce the interference of reflection. We minimized other interferences of inhomogeneous color intensity by maintaining an equal condition in each measurement.
Figure 2(a) was updated.
We have also revised the Methods section to clarify the procedure.
- Is the ΔRGB calculating value include only blue color coordinate or three color coordinates?
Response:
Thank you for the comment.
The ΔRGB value calculated the red, green, and blue coordinates. This was a typographical error, and we have revised it in the Methods section.
Reviewer 2 Report
This work introduces a paper-based analytical device for paraquat in clinical samples. Paraquat, as a widely used herbicide, has the possibility of causing some disorders. Therefore, its measurement is in high demand. The proposed strategy is interesting but lacks various scientific data. I have meticulously reviewed this work and do not recommend it as a publication in Diagnostics Journal.
My comments are as below:
1- I suggest that the Authors implement the state of the art in the Introduction, better describing how their strategy is collocated within the current scenario. They could introduce about paper-based analytical devices (PADs) and recent progresses as well (some parts of mentioned explanations in Discussion section about PADs can be moved here).
2- Some typos such as “The Schematic” (page 3 line 101) should be corrected.
3- The manuscript lacks in-depth discussion to relate each data. For example, the signal readout method has not been addressed appropriately. How did the Authors overcome to the ambient light effect? Settings and brand of the utilized camera should be reported.
4- Regarding to the image processing, how did the Authors select region of interest (ROI)? Which software (version and developer) and algorithm have been used? This should be addressed properly.
5- Device fabrication process should be accurately reported.
6- How is the stability if the device? How long would it be possible to store and use it? The reproducibility should be reported.
7- I recommend the Authors to submit a statement on ethical approval, along with the full name of the committee for this study because urine samples have been utilized in this study.
8- Conclusion section should be thoroughly revised.
Author Response
We greatly appreciate the time and effort the editor and reviewers have put into our paper. Below we have outlined our responses to the reviewer’s comments point by point. We hope that the following responses and the corresponding revision of the manuscript meet the editor’s and reviewers’ requirements for considering this manuscript for publication in Diagnostics.
In the following sections, we have restated the reviewer’s comments followed by our response to each comment.
- I suggest that the Authors implement the state of the art in the Introduction, better describing how their strategy is collocated within the current scenario. They could introduce about paper-based analytical devices (PADs) and recent progresses as well (some parts of mentioned explanations in Discussion section about PADs can be moved here)?
Response:
Thank you for the important comment.
We have introduced the concept of the point-of-care test (POCT) and the current application of paper-based analytical devices in the Introduction section.
- Some typos such as “The Schematic” (page 3 line 101) should be corrected..
Response:
Thank you for the comment. We have corrected this.
- The manuscript lacks in-depth discussion to relate each data. For example, the signal readout method has not been addressed appropriately. How did the Authors overcome to the ambient light effect? Settings and brand of the utilized camera should be reported.
Response:
Thank you for the comment.
The reaction in the paper-based analytical device was captured using a digital camera perpendicularly in ambient light. The digital camera (EOS 5D Mark III, Canon, Japan) was set in auto-focus mode.
Then, we analyzed the color intensity change using Image J software. The region of interest (ROI) of each well was selected as a circle with a diameter of 0.38 cm (95% of the diameter of each well) to reduce the interference of reflection. We minimized other interferences of inhomogeneous color intensity by maintaining an equal condition in each measurement.
The Methods section has been revised accordingly.
- Regarding to the image processing, how did the Authors select region of interest (ROI)? Which software (version and developer) and algorithm have been used? This should be addressed properly..
Response:
Thank you for the comment.
The resulting data were analyzed using ImageJ software (Version 2.0.0, National Institute of Health). The region of interest (ROI) of each well was selected as a circle with a diameter of 0.38 cm (95% of the diameter of each well) to reduce the interference of reflection. We minimized other interferences of inhomogeneous color intensity by maintaining an equal condition in each measurement.
The Methods section has been revised accordingly.
- Device fabrication process should be accurately reported.
Response:
Thank you for the important comment.
We have carefully revised the Methods section to describe the fabrication process of our paper-based analytical device in detail. The schematic image has also been updated.
- How is the stability if the device? How long would it be possible to store and use it? The reproducibility should be reported.
Response:
Thank you for the pertinent comment.
Regarding reproducibility, we have started an engineering improvement program to address this important issue, and it would take another year. In our current study, an interesting idea was proposed, and a practice protocol for the quick diagnosis of paraquat in urine was developed
- I recommend the Authors to submit a statement on ethical approval, along with the full name of the committee for this study because urine samples have been utilized in this study.
Response:
We have submitted the Institutional Review Board approval documentation.
- Conclusion section should be thoroughly revised.
Response:
We sincerely appreciate this suggestion.
We have carefully revised this section.
Reviewer 3 Report
Authors developed a paper-based microfluidic device for the detection of paraquat in human urine. Paraquat is a chemical herbicide, or weed killer, that's highly toxic and can cause fatal poisoning when ingested or inhaled. It is very useful to have a low-cost and robust method to detect this chemical in a rapid manner. I believe the manuscript has potential to be published in Diagnostics, however; it requires some changes before acceptance.
- What is the maximum residue limit of paraquat in food?
- What is the normal range of paraquat in human urine?
- The limit of detection and dynamic range should be mentioned in the abstract.
- The limit of detection of the device should be compared with the maximum residue limit of paraquat in urine to see if the device is useful.
- For the paper-based microfluidic device, they have captured the results by a digital camera before measuring the color intensity of the developed colors in the detection zones. As shown in Figure 2a, they have captured the detection zones when they were completely wet. However, it is not a proper way to do this part of the experiment, the light reflection in the detection zones avoids the correct measuring of the developed color intensities by ImageJ. The authors should repeat the experiment and let the detection zones dry for around 10 min before they can capture them.
Author Response
- 1. What is the maximum residue limit of paraquat in food?
Response:
The maximum residue limit of paraquat in food is around 0.01–0.2 ppm.
Reference :
- FAO/WHO codex alimentarius. Available online: http://www.fao.org/fao-who-codexalimentarius/codex-texts/dbs/pestres/pesticide-detail/en/?p_id=57 (accessed on 29/11/2020).
- What is the normal range of paraquat in human urine?
Response:
Thank you for this important comment. According to the World Health Organization WHO /PCS hazard classification (WHO 2002), paraquat dichloride is moderately hazardous (class II). There is no regulation that defines the normal range of paraquat in human urine. The acute reference dose of paraquat (ARfD) is 0.005 mg/kg body weight, and the acceptable daily intake (ADI) is 0.004 mg/kg body weight/day.
Reference :
- FAO SPECIFICATIONS AND EVALUATIONS FOR PARAQUAT DICHLORIDE. Available online: http://www.fao.org/fileadmin/templates/agphome/documents/Pests_Pesticides/Specs/Paraquat08.pdf (accessed on 29/11/2020).
- Available online: https://pubchem.ncbi.nlm.nih.gov/compound/Paraquat (accessed on 29/11/2020).
3.The limit of detection and dynamic range should be mentioned in the abstract.
Response:
Thank you for the important comment.
The R2 value for urine paraquat standard curve was 0.9989, with a dynamic range of 0–100 ppm. The limit of detection was 3.00688 ppm.
The abstract has been revised accordingly.
- The limit of detection of the device should be compared with the maximum residue limit of paraquat in urine to see if the device is useful.
Response:
The limit of paraquat detection in our system was 3.00688 ppm. Patients with this urine concentration or higher may develop serious clinical symptoms and have a poor prognosis. According to previous studies, patients with semiquantitative urine test results showing more than navy blue ( >10 ppm) in the initial 24 h have a high probability of death [1,2]. A recent study suggested that an initial urine paraquat concentration over 32.2 ppm had a significantly low 28-day survival rate [3].
There is no regulation that defines the maximum residue limit of paraquat in urine. The paraquat acceptable daily intake (ADI) is 0.004 mg/kg body weight/day [4]. When comparing with the paraquat ADI, our paper-based analytical device may not detect low-dose paraquat exposure.
Our paper-based analytical device was developed as a point-of-care test. We focused on acute paraquat intoxication and follow-up. We did not aim to replace time-consuming, sophisticated measurements. Furthermore, using our paper-based analytical device, clinically significant paraquat intoxication in patients could be accurately detected much earlier than the results of the definite measurement. The time is precious for emergent hemoperfusion. In addition, increasing availability could improve the diagnostic rate in urban areas.
Moreover, our paper-based analytical device is not designed for paraquat exposure surveillance, which requires higher resolution. We could not use our paper-based analytical device to exclude paraquat exposure.
Therefore, our paper-based analytical device is an ideal point-of-care tool that is demonstrably accurate, less invasive, easy-to-use, and time- and cost-effective. This device is useful for acute paraquat intoxication management
Reference :
- Scherrmann, J.M.; Houze, P.; Bismuth, C.; Bourdon, R. Prognostic value of plasma and urine paraquat concentration. Hum Toxicol 1987, 6, 91-93, doi:10.1177/096032718700600116.
- Dinis-Oliveira, R.J.; Duarte, J.A.; Sanchez-Navarro, A.; Remiao, F.; Bastos, M.L.; Carvalho, F. Paraquat poisonings: mechanisms of lung toxicity, clinical features, and treatment. Crit Rev Toxicol 2008, 38, 13-71, doi:10.1080/10408440701669959
- Liu, X.W.; Ma, T.; Li, L.L.; Qu, B.; Liu, Z. Predictive values of urine paraquat concentration, dose of poison, arterial blood lactate and APACHE II score in the prognosis of patients with acute paraquat poisoning. Exp Ther Med 2017, 14, 79-86, doi:10.3892/etm.2017.4463.
- Available online: https://pubchem.ncbi.nlm.nih.gov/compound/Paraquat (accessed on 29/11/2020).
- For the paper-based microfluidic device, they have captured the results by a digital camera before measuring the color intensity of the developed colors in the detection zones. As shown in Figure 2a, they have captured the detection zones when they were completely wet. However, it is not a proper way to do this part of the experiment, the light reflection in the detection zones avoids the correct measuring of the developed color intensities by ImageJ. The authors should repeat the experiment and let the detection zones dry for around 10 min before they can capture them.
Response:
Thank you for the critical comment.
Our reaction time was set at 10 min according to the results of kinetic studies.
We conducted kinetic studies for the developed paper-based device. We set the reaction time to 0, 5, 15, and 20 min. We measured the ΔRGB value in the same manner used for the experimental protocol, except for the reaction time. The results of the kinetic studies are presented in Table S1 and Figure S1. At a reaction time of 10 min, the highest ΔRGB value was obtained. Different paraquat concentrations presented the same trend in kinetics studies: the ΔRGB value reached the maximum level at 10 min of reaction time. As an example, we present the measurement results for 50 ppm paraquat.
To obtain better detection results, we did not let the detection zone dry and captured the image at 10 min. To deal with reflection and other uneven color intensities, we performed several steps to reduce interference. First, we captured the image perpendicularly under the same conditions for all images. Then, the region of interest (ROI) of each well was selected as a circle with a diameter of 0.38 cm (95% of the diameter of each well), (the greatest reflection occurred in the surrounding barrier of each well). We reduced the other interferences of inhomogeneous color intensity by maintaining an equal condition in each measurement.
Table S1 Developed paper-based device measurement results for 50 ppm paraquat at different reaction times
|
Delta RGB* |
0 min |
5 min |
10 min |
15 min |
20 min |
|
Test 1 |
48.3559 |
63.5782 |
81.6701 |
50.6486 |
61.3760 |
|
Test 2 |
37.1812 |
57.5128 |
80.0336 |
48.1173 |
55.7682 |
|
Test 3 |
49.4987 |
69.5623 |
80.9990 |
54.0720 |
64.8358 |
|
Average |
45.0119 |
63.5511 |
80.9009 |
50.9460 |
60.6600 |
|
Standard |
6.8056 |
6.0248 |
0.8227 |
2.9884 |
4.5760 |

Round 2
Reviewer 2 Report
The revised version provides is now easily understandable for the readers but still a minor revision is required.
1- In Abstract line 28, the R2 should be replaced with R2.
“The R2 value for urine paraquat standard…”.
2- LoD and LoQ values should be revised in the Table 1 and accordingly in whole text:
LoD 3.00688 -----> 3.01
LOQ 10,0229 -----> 10.02
Author Response
We sincerely appreciate your comment. We have revised the manuscript and re-submitted our revised manuscript. Thank you very much!